# Genomic Features of Antimicrobial Resistance in *Staphylococcus pseudintermedius* Isolated from Dogs with Pyoderma in Argentina and the United States: A Comparative Study

**DOI:** 10.3390/ijms241411361

**Published:** 2023-07-12

**Authors:** Mariela E. Srednik, Claudia A. Perea, Gabriela I. Giacoboni, Jessica A. Hicks, Christine L. Foxx, Beth Harris, Linda K. Schlater

**Affiliations:** 1Postdoctoral Research Participation Program, Office of Research in Science and Education, Oak Ridge Associated Universities, Oak Ridge, TN 37831, USA; 2Diagnostic Bacteriology and Pathology Laboratory, National Veterinary Services Laboratories, Animal and Plant Health Inspection Service, United States Department of Agriculture, Ames, IA 50010, USA; 3Departamento de Microbiología, Facultad de Ciencias Veterinarias, Universidad Nacional de La Plata, La Plata 1427, Argentina; 4Transboundary Disease Analytics, Center for Epidemiology and Animal Health, Veterinary Services, Animal and Plant Health Inspection Service, United States Department of Agriculture, Fort Collins, CO 80526, USA; 5National Animal Health Laboratory Network, National Veterinary Services Laboratories, Animal and Plant Health Inspection Service, United States Department of Agriculture, Ames, IA 50010, USA

**Keywords:** *Staphylococcus pseudintermedius*, methicillin resistance, multi-drug resistance, SCC*mec*, clonal relationship, phylogeny, canine pyoderma

## Abstract

*Staphylococcus pseudintermedius* is the most common opportunistic pathogen in dogs and methicillin resistance (MRSP) has been identified as an emerging problem in canine pyoderma. Here, we evaluated the antimicrobial resistance (AMR) features and phylogeny of *S. pseudintermedius* isolated from canine pyoderma cases in Argentina (*n* = 29) and the United States (*n* = 29). 62% of isolates showed multi-drug resistance. The AMR genes found: *mec*A, *bla*Z, *ermB*, *dfr*G, *cat*A, *tet*M, *aac*(6′)-*aph*(2″), in addition to *tet*K and *lnu*A (only found in U.S. isolates). Two point mutations were detected: *grl*A(S80I)-*gyr*A(S84L), and *grl*A(D84N)-*gyr*A(S84L) in one U.S. isolate. A mutation in *rpo*B (H481N) was found in two isolates from Argentina. SCC*mec* type III, SCC*mec* type V, ΨSCC*mec*_57395_ were identified in the Argentinian isolates; and SCC*mec* type III, SCC*mec* type IVg, SCC*mec* type V, and SCC*mec* type VII variant in the U.S. cohort. Sequence type (ST) ST71 belonging to a dominant clone was found in isolates from both countries, and ST45 only in Argentinian isolates. This is the first study to comparatively analyze the population structure of canine pyoderma-associated *S. pseudintermedius* isolates in Argentina and in the U.S. It is important to maintain surveillance on *S. pseudintermedius* populations to monitor AMR and gain further understanding of its evolution and dissemination.

## 1. Introduction

*Staphylococcus pseudintermedius* is the most common opportunistic pathogen found in dogs and is part of their normal skin microbiota [1]. However, when the skin barrier or immune system becomes compromised*, S. pseudintermedius* can cause pyoderma, urinary infections, and otitis externa [1]. Methicillin-resistant *S. pseudintermedius* (MRSP) emerged in dogs in the 2000s; the prevalence of MRSP colonization in dogs depends on the geographic and clinical characteristics of the population under study [2]. Companion animals play an important role in the epidemiology of MRSP, as staphylococci can be transmitted to humans easily via close contact [3]. Consequently, humans can also become transient carriers of MRSP after contact with their colonized dogs [3], leading to severe disease in clinical populations [4]. The first MRSP infections in humans were recorded in 2006 [5], and the first MRSP isolated from a human patient in Argentina was in 2020 [6]. Treatment options for MRSP infections represent a new challenge in veterinary medicine, and infections in humans are often underreported due to inaccurate identification as *S. aureus* [7,8].

Methicillin resistance in *S. pseudintermedius* is predominantly due to the presence of the *mec*A gene, which encodes a protein (PBP2a) that has low affinity for β-lactam antibiotics. The *mec*A gene can be found in a mobile element of the bacterial chromosome known as the staphylococcal chromosomal cassette (SCC*mec*). SCC*mec* elements are characterized by the integrity of the methicillin resistance regulon containing the *mec*A gene, the allotype of the recombinase genes, and the general genetic structure [9]. To date, 14 SCC*mec* types (I–XIV) and several subtypes have been described [10,11]. In addition, many novel SCC*mec* elements have been described in *S. pseudintermedius* [12]. Adding to its importance, SCC*mec* can be transferred between different species of *Staphylococcus* [13].

The incidence of MRSP has increased significantly worldwide and is concerning due to concomitant increases in detection of multi-drug resistance (MDR; defined here as acquired non-susceptibility to at least one agent in three or more antimicrobial classes [14]), limited treatment options for infections caused by this organism, and the potential risk of zoonotic transmission from animals to humans. Antimicrobial resistance (AMR) is considered one of the most serious global health threats to both animals and humans. Staphylococcal penicillin resistance is very common and typically conferred by a β-lactamase encoded by the *bla*Z gene [15]. *S. pseudintermedius* isolates carrying SCC*mec* cassettes are not only resistant to penicillin and other β-lactams, such as cephalosporins and carbapenems, but often also possess resistance to macrolides, lincosamides, and streptogramins (MLS). These data are supported by the detection of several genes conferring resistance to MLS in staphylococci: *erm*A, *erm*B, and *erm*C genes confer resistance by targeted modification of ribosomal RNA (inducible or constitutive); the *msr*A gene confers resistance to macrolides and streptogramins by mediating drug efflux; and the *lnu*A gene confers resistance to lincosamides by direct inactivation [16].

Antimicrobial use can lead to co-selection and emergence of resistant strains, such as MRSP, through the acquisition of mobile genetic elements and/or mutations [17]. Clonal expansion and geographical dissemination of certain sequence types (ST) have recently been associated with the horizontal acquisition of AMR genes, leading to a global emergence of multi-drug-resistant MRSP clones [18]. Multi-locus sequence typing has identified several predominant MRSP clones around the world, including ST71 in Europe, ST68 in the United States (U.S.), and ST45/ST112 in Asia [19]. Additionally, there is evidence of clonal expansion of MRSP lineages that have disseminated over large distances [20]. Despite MRSP’s clinical importance and the number of studies on genotype-to-phenotype relationships for AMR in a variety of veterinary pathogens, few publications related to AMR in *S. pseudintermedius* examine clonal ST with AMR phenotype, genotype, and SCC*mec* relationships at a population scale. We aim to fill this gap in the literature by deeply characterizing and comparing the phenotypic and genotypic profiles of MRSP clones from dogs with pyoderma banked at the Universidad Nacional de la Plata in Argentina and at the Animal and Plant Health Inspection Service’s National Veterinary Services Laboratories (NVSL) in the United States, as part of the National Animal Health Laboratory Network (NAHLN) Antimicrobial Resistance Pilot Project.

To reach this level of comparative characterization, phenotypic data for MRSP clones recovered from canine pyoderma cases in Argentina and the U.S. were obtained by testing against antimicrobial drug panels and interpreted as sensitive, intermediate, or resistance using breakpoint guidelines published in the 2020 *Vet01S* Clinical and Laboratory Standards Institute [21]. Genotypic data for the same MRSP clones were also obtained using untargeted whole-genome sequencing and in silico detection of AMR genes (AMRFinder Plus, ABRicate, and ResFinder), SCC*mec* types and subtypes (SCC*mec*Finder), and multi-locus sequence types (MLST and ABRicate). Phylogenetic reconstruction using single-nucleotide polymorphism (SNP) data was used to generate dendrograms and heatmaps, which alongside genotype-to-phenotype correlation tables were used to identify population-dependent trends in phenotype-to-genotype concordance and discordance.

## 2. Results

### 2.1. Antimicrobial Susceptibility Testing

All isolates from Argentina (*n* = 29) and the U.S. (*n* = 29) were susceptible to nitrofurantoin, imipenem, and vancomycin.

All β-lactams (penicillin, ampicillin, oxacillin, cephalothin, cefazolin, cefovecin, cefpodoxime, imipenem) were grouped as one antimicrobial class. Twenty-four isolates (82.7%) from Argentina showed resistance to at least three antimicrobial classes and were classified as MDR (Figure 1). One methicillin-susceptible *S. pseudintermedius* (MSSP) and all 23 MRSP isolates from Argentina were MDR, compared to only 12 (*n* = 3 MSSP, *n* = 9 MRSP) or 41.4% of isolates from the U.S. A greater proportion of isolates from Argentina showed resistance to erythromycin (79.3%), clindamycin (79.3%), fluoroquinolones (68.9%), chloramphenicol (31.0%), trimethoprim/sulfamethoxazole (82.7%) and rifampin (6.9%) compared to the U.S. isolates. On the other hand, a higher proportion of isolates from the U.S. showed resistance to tetracyclines (37.9%) and gentamycin (17.2%).

For β-lactams, 72.4% (*n* = 21) of the Argentine isolates were oxacillin-resistant; this largely aligned with genotypic data demonstrating that 79.3% (*n* = 23) of the Argentine isolates harbored the *mec*A gene and were classified as MRSP. Among the U.S. isolates, 44.8% (*n* = 13) were oxacillin-resistant and carried the *mec*A gene.

Phenotypic resistance to penicillin was detected in 27 (93.1%) isolates from Argentina and 25 (86.2%) isolates from the U.S. The distribution of MRSP and MSSP penicillin-resistant isolates were markedly different between the two countries: in Argentina, 73.9% (*n* = 17) of penicillin-resistant MRSP isolates carried the *bla*Z gene; 66.7% (*n* = 4) MSSP isolates carried the *bla*Z gene and showed concordant penicillin resistance. In the U.S., 84.6% (*n* = 11) of the MRSP isolates harbored the *bla*Z gene and showed penicillin resistance; only 16.7% (*n* = 2) of MSSP isolates with penicillin resistant phenotypes also carried the *bla*Z gene. Of note, four MSSP isolates had penicillin-sensitive phenotypes despite *mec*A-dependent oxacillin-resistant genotypes. In total, 21 MRSP isolates from Argentina and 13 MRSP isolates from the U.S. were penicillin resistant and had concordant genotypes conferring β-lactam resistance (*bla*Z and *mec*A detection). Interestingly, the β-lactam response regulator genes *bla*R1 and *bla*I were detected in a penicillin-resistant isolate from the U.S. in lieu of *bla*Z.

Resistance to macrolides, lincosamides, and streptogramines (MLS) was detected in 79.3% (*n* = 23) of all Argentine isolates, of which the concordant MLS resistance conferring *erm*B gene was detected in only 15 isolates (65.2%). Among the U.S. isolates, only 41.4% (*n* = 12) were resistant to MLS, of which all (100.0%) carried the concordant *erm*B gene.

For aminoglycosides, amikacin interpretation using the Sensititre™ plate was not possible (MIC ≤ 16 μg/mL). Three isolates from Argentina showed resistance to gentamicin, but the gene that encodes the bi-functional enzyme *aac*(6′)-*aph*(2″)-Ia conferring resistance to amikacin, gentamicin, kanamycin and tobramycin was detected with partial coverage (≤60% of the target gene) in six isolates. Five isolates from the U.S. showed resistance and five additional isolates showed intermediate resistance to gentamicin, for which the *aac*(6′)-*aph*(2″)-Ia gene was detected, but four with only partial coverage. Interestingly, one isolate that presented the gene was phenotypically susceptible.

Twenty isolates (68.9%) from Argentina were phenotypically resistant to fluoroquinolones (enrofloxacin, marbofloxacin, pradofloxacin) and in all these fluoroquinolone resistant isolates two point mutations were detected: *grl*A(S80I)-*gyr*A(S84L). Four isolates showed intermediate susceptibility to enrofloxacin, susceptibility to marbofloxacin and pradofloxacin, and only presented a single point mutation in *grl*A(S80I). Seven isolates (24.1%) from the U.S. were phenotypically resistant to fluoroquinolones and presented the aforementioned point mutations; however, one fluoroquinolone resistant isolate presented intermediate susceptibility to pradofloxacin and had a variant mutation for the *grl*A gene (D84N). Two isolates showed intermediate susceptibility to enrofloxacin and susceptibility to the other fluoroquinolones tested, and presented the *grl*A(S80I) mutation only.

For phenicols, phenotypic resistance to chloramphenicol was found in 31.0% (*n* = 9) of the isolates from Argentina, of which 33.3% (*n* = 3) lacked the concordant chloramphenicol resistance conferring *cat*A gene. In the U.S. sample, 100.0% (*n* = 4*)* of chloramphenicol-resistant isolates also carried the *cat*A gene.

For folate pathway antagonists, 82.7% (*n* = 24) of the isolates from Argentina and 31.0% (*n* = 9) of isolates from the U.S. were resistant to trimethoprim/sulfamethoxazole; all trimethoprim/sulfamethoxazole-resistant isolates, regardless of the study sample source, harbored the concordant resistance-conferring *dfr*G gene.

Resistance to rifampin was found in two Argentine isolates, and it was associated with a nonsynonymous mutation in the *rpo*B gene, corresponding to residue 481 (H481N) structural alteration from histidine to asparagine.

Figure 2 shows the phenotypic AMR profiles for all isolates included in this study, clustered based on the similarity of resistance interpretations for each antimicrobial class as described in the Methods. Interestingly, the clustering showed concordance with MLST profiles ST1412, ST2257, ST71 and ST339. No other significant associations were found (i.e., country of origin). Additionally, discrepancies between AMR phenotype (classified as binary-coded resistant/susceptible or intermediate) and concordant genotype can be visualized as they have been described in the Results. For clindamycin and erythromycin, for example, a total of eight isolates shows a resistant phenotype but lack the corresponding gene associated with resistance to these antimicrobials.

### 2.2. Population Structure

A total of 24 sequence types (STs) were identified by multi-locus sequence typing (MLST) for the *S. pseudintermedius* isolates from Argentina, five of which have been previously described (Table 1): ST339 (*n* = 7), ST1412 (*n* = 3), ST71 (*n* = 2), ST45 (*n* = 1) and ST313 (*n* = 1); and 15 newly identified STs from this study: ST2233–2244 and ST2259–2261. Among isolates from the U.S., a total of 29 STs were identified, 11 described previously (Table 1): ST71, ST97, ST181, ST301, ST440, ST551, ST1055, ST1229, ST1420, ST1431, ST1692, and 17 STs first described here: ST2245–2258 and 2262–2264.

To determine the clonal relationships between the STs identified in this study, a minimum spanning tree (Figure 3) was generated using the *goe*BURST algorithm in PHYLOViZ (http://phyloviz.net/, accessed on 12 May 2022). A clonal complex (CC) consisted of allelic profiles with five or more allele matches, while singletons were unrelated to any other within this dataset. In this case, six CCs and 21 singletons were identified: the largest CC was composed of 14 STs differing by two loci, where only ST339 and ST313 are single-locus variants from one another. The second largest CC consisted of four newly identified STs in this study (ST2234, ST2235, ST2233 and ST2242), also with a difference of two loci. The four remaining CCs consisted of only two STs each, where again, the double-locus variant dominates. All isolates that shared the same sequence type (ST339, ST1412, and ST71) were susceptible to tetracyclines and resistant to at least one antibiotic from each of the five antimicrobial classes (β-lactams, macrolides, lincosamides, fluoroquinolones, trimethoprim/sulfamethoxazole). Additional shared phenotypic resistance paralleling ST included resistance to chloramphenicol among the three ST1412 isolates. Among the three ST71 isolates, the two from Argentina presented resistance to rifampin, whereas the ST71 isolate from the U.S. was susceptible.

The maximum likelihood phylogeny (Figure 4) based on the concatenated sequences of single-nucleotide polymorphisms (SNPs) revealed deep branching in almost all the isolates, indicating a high degree of diversity based on a colloquial visual observation in branching structure among the study population. Clades corresponding to ST71, ST1412, and ST339 showed more homogeneity among the isolates, possibly due to clonal expansion. Regarding geographical origin, most of the isolates from Argentina fell within one of the three main clades observed, whereas the isolates from the U.S. make up the other two. Consequently, the U.S. isolates are more closely related to each other than to isolates from Argentina, and vice versa. However, there were three isolates from Argentina and one isolate from the United States that fell within the others’ predominant clades. Closer inspection shows that two Argentine isolates in this category had greater genetic relatedness to a U.S. isolate and were all typed ST71. The high degree of diversity is also supported by the proportion of unique STs identified in relation to the total number of isolates studied: 47 STs out of 58 isolates (81.0%). Additionally, pairwise distances from 921 to 20,237 SNPs between isolates was observed. Interestingly, the isolates that belong to the two dominant MRSP clones (ST71 and ST45) fall within the same clade.

### 2.3. Antimicrobial Resistance Genes

The most common AMR genes identified among this cohort of *S. pseudintermedius* isolates were *mec*A and *bla*Z for β-lactams, *erm*B for macrolides, *lnu*A for lincosamides, *tet*M and *tet*K for tetracyclines, *aac*(6′)-*aph*(2″), *aph*(3′)-IIIa, *ant*(3)-la, and *sat*4 for aminoglycosides, *cat*A for phenicols, and *dfr*G for folate pathway antagonists, which are shown in Figure 4. Detection of the *mec*A gene was used to determine 23 MRSP isolates from Argentina and 13 MRSP isolates from the U.S., respectively. Among these, 47.8% (*n* = 11) of the isolates from Argentina and 46.2% (*n* = 6) of the isolates from the U.S. also carried one or both *mec*A regulator genes, *mec*I and *mec*R1.

#### 2.3.1. MRSP

Among MRSP isolates, the most frequent antimicrobial resistant genes detected were those associated to aminoglycoside resistance, such as the aminoglycoside phosphotransferase gene *aph*(3′)-IIIa, the aminoglycoside nucleotidyltransferase gene *ant*(6)-la, and the streptothricin acetyltransferase *sat*4 gene, as well as the *dfr*G gene associated with resistance to trimethoprim-sulfamethoxazole, at proportions of 95.6% (*n* = 22) for Argentina and 69.2% (*n* = 9) for the U.S.. Additionally, the bifunctional acetyltransferase/phosphotransferase gene *aac*(6′)-*aph*(2′)-la associated with resistance to gentamicin and other aminoglycosides was detected in 26.1% (*n* = 6) and 76.9% (*n* = 10) of MRSP isolates from Argentina and the U.S., respectively.

The *bla*Z gene associated with penicillin resistance was detected in 73.9% (*n* = 17) of MRSP isolates from Argentina and 84.6% (*n* = 11) of MRSP isolates from the U.S. The *erm*B gene associated with MLS resistance was detected in 65.2% (*n* = 15) of Argentine MRSP isolates and 69.2% (*n* = 9) of U.S. MRSP isolates. Tetracycline resistant genotypes were identified in five Argentine MRSP isolates (21.7% *tet*M detections); conversely, tetracycline resistant genotypes were identified in seven U.S. MRSP isolates (38.5% *tet*M detections, 15.4% *tet*M and *tet*K detections). Resistance to chloramphenicol was associated with the presence of the *cat*A gene, which encodes a chloramphenicol acetyltransferase, in 26.1% (*n* = 6) of Argentine MRSP isolates and 15.4% (*n* = 2) of U.S. MRSP isolates. Resistance to fluoroquinolones was associated with two point mutations: *grl*A(S80I) and *gyr*A(S84L) in 82.6% (*n* = 19) of Argentine MRSP isolates and in 46.2% (*n* = 6) of U.S. MRSP isolates. Interestingly, one isolate from the U.S. presented a variant mutation in *grl*A from S80I to D84N. Four MRSP isolates from Argentina had a single point mutation corresponding to *grl*A(S80I) and four isolates from the U.S. had a single point mutation in *grl*A (*n* = 1 D84Y, *n* = 1 S80R, and *n* = 2 S80I).

#### 2.3.2. MSSP

Among MSSP isolates, 66.7% (*n* = 4) of Argentine isolates harbored the *bla*Z gene, compared to 81.2% (*n* = 13) of U.S. isolates. Only one MSSP isolate from Argentina was phenotypically characterized as MDR, with concomitant resistance gene detections associated with penicillin resistance (*bla*Z), aminoglycoside resistance (*aph*(3′)-IIIa, *ant*(6′)-la and *sat*4), trimethoprim-sulfamethoxazole resistance (*dfr*G), and two point mutations in *grl*A(S80I) and *gyr*A(S84L) associated with fluoroquinolone resistance. Among MSSP isolates from the U.S., the *tet*M gene was found in 25% (*n* = 4) of isolates, *erm*B in 18.7% (*n* = 3) of isolates, and *cat*A, *aph*(3′)-IIIa, a*nt*(6)-la and *sat*4 in 12.5% (n = 2) of isolates. In addition, the *lnu*A gene was detected once in only one U.S. isolate. No MSSP isolates from the U.S. were identified as phenotypically MDR, and no mutations in *grl*A and *gyr*A were detected.

Phenotype–genotype resistance correlations for sixteen antimicrobials are shown in Table 2. One hundred percent phenotype–genotype concordance was observed for trimethoprim-sulfamethoxazole, fluoroquinolones, tetracyclines, rifampin, nitrofurantoin and vancomycin. Lower correlations were observed for gentamicin, where nine isolates were phenotypically susceptible or intermediate but harbored the bifunctional *aac*(6′)-*aph*(2″)-Ia gene, followed by erythromycin and clindamycin, with eight Argentine isolates tested to be phenotypically resistant in the absence of any associated resistance genes. Resistance genotypes and phenotypes overlapped at 96.5% concordance for oxacillin, with two Argentine isolates harboring the *mec*A gene but with a susceptible phenotype (confirmed at UNLP by disk diffusion tests using oxacillin). Regarding penicillin, two isolates were phenotypically susceptible but harbored the *bla*Z gene, and one isolate was phenotypically resistant but lacked any β-lactam-associated resistance genes. For chloramphenicol, three isolates were phenotypically resistant but there was no detection of a resistance gene.

### 2.4. SCCmec Types

SCC*mec*Finder was used to classify the MRSP isolates from Argentina into two SCC*mec* types: SCC*mec* type III (*n* = 2) and SCC*mec* type V (*n* = 10). Of the 10 Argentine isolates identified as SCC*mec* type V, three were classified as type V(5C2), with only one *ccr*C1 recombinase, and seven were type V(5C2&5), which corresponds to two *ccr*C1 recombinase allotypes. The 13 MRSP isolates from the U.S. were classified into four SCC*mec* types: SCC*mec* type III (*n* = 2), SCC*mec* type IVg (*n* = 4), SCC*mec* type V (*n* = 6), and SCC*mec* type VII (*n* = 1). Similarly, of the SCC*mec* type V-classified U.S. isolates, one was further characterized as type V(5C2) and three were characterized as type V(5C2&5). Table 3 summarizes the characteristics of the SCC*mec* elements found in this study.

SCC*mec*Finder is designed to detect all the reference SCC*mec* types listed by the International Working Group on the Classification of Staphylococcal Cassette Chromosome Elements (IWG-SCC) in short read sequences. However, the tool was originally built for *S. aureus* sequence typing. For this reason, the SCC*mec* cassettes predicted by SCC*mec*Finder for *S. pseudintermedius* isolates in this study were additionally mapped against known reference sequences, for which similarity (homology) and coverage values (%) were compared to obtain a “best match” (Appendix A). As an example, isolates predicted as SCC*mec* type V (5C2&5) had sequences mapped against all SCC*mec* type V references. The resultant search demonstrated that isolates with SCC*mec* type V (5C2&5) had high homology (84.8–99.9%) with *S. pseudintermedius* 06-3228 (FJ544922.1) and, to a lesser degree, *S. pseudintermedius* 23929 (ERR175868; Figure 5A). SCC*mec* type V (5C2&5) was also previously described as SCC*mec* V_T_, a truncated version of the SCC*mec* VII (AB462393) with two *ccr*C1 recombinases (*ccr*C2 and *ccr*C8). Four isolates harbored the *czr*C gene present in SCC*mec* type Vc and absent in SCC*mec* type Vb (Figure 5B,C). All *S. pseudintermedius* isolates predicted as SCC*mec* type V(5C2) harbored one *ccr*C1 complex and shared the highest homology with *S. aureus* reference AB121219.1, classified as SCC*mec* Va (Figure 5A).

Four isolates were classified as SCC*mec* type III and shared the greatest homology with *S. pseudintermedius* KM1381 (AM904732.1), which harbors a hybrid SCC*mec* type II-III (Figure 5D). Four isolates from the U.S. were classified as SCC*mec* IVg and were mapped against all SCC*mec* type IV references, obtaining higher homology with *S. aureus* SCC*mec* IVg reference (Figure 5E). One SCC*mec* type VII-classified U.S. isolate was mapped against two references: SCC*mec* VII reference *S. aureus* JCSC6082 (AB373032.1) and SCC*mec* VII variant reference *S. pseudintermedius* NA45 (CP016072.1, also known as SCC*mec*_NA45_), the latter of which shared the greatest homology (98.70%) with the queried isolate. Interestingly, while the *mec*A gene in this isolate presented the same orientation as did in the SCC*mec*_NA45_ reference genome, the *ccr*C6 recombinase was juxtaposed to that in SCC*mec*_NA45_ (Figure 5F). Finally, one isolate from Argentina belonging to ST45 harbored a ΨSCC*mec*_57395_ cassette.

## 3. Discussion

In this study, we analyzed *S. pseudintermedius* isolates from the Buenos Aires Metropolitan Area of Argentina and compared them to isolates obtained from different states of the U.S. We observed differences in the resistance profiles and genomic features within and between the two groups. *S. pseudintermedius* isolates are commonly resistant to the most frequently used antimicrobials in dogs, such as penicillins, tetracyclines and macrolides [22,23].

In this study, MRSP isolates showed resistance to trimethoprim/sulfamethoxazole, erythromycin and clindamycin, fluoroquinolones, chloramphenicol, tetracycline, and gentamicin. Resistance to rifampin was observed only in isolates from Argentina. All MRSP isolates from Argentina had a MDR profile showing resistance mainly to penicillin (100%), macrolides and lincosamides (95.7%), trimethoprim-sulfamethoxazole (95.7%), fluoroquinolones, (82.6%), chloramphenicol (39.1%), tetracyclines (21.7%), gentamycin (13%), and rifampin (8.7%). We observed differences in the U.S. resistance profiles, only 69.2% isolates in the MRSP group were MDR, which included resistance mainly to penicillin (100%), macrolides and lincosamides (69.2%), trimethoprim/sulfamethoxazole (69.2%), tetracyclines (61.5%), and fluoroquinolones (61.5%), gentamycin (30.7%), and chloramphenicol (15.4%). In this regard, resistance to tetracycline in isolates from Argentina was low compared to isolates from the U.S.; other studies have also reported this low tetracycline resistance seen in Argentine isolates [22,23].

For *S. pseudintermedius*, the screening test for methicillin resistance using cefoxitin disk diffusion testing (used for *S. aureus*) leads to an unacceptably high percentage of false-negative results and has been reported to be inappropriate [24]. For this reason, oxacillin MIC or oxacillin disk are recommended for use in dogs. Nevertheless, the most reliable test for the detection of methicillin resistance is *mec*A PCR; however, few laboratories perform PCR for *mec*A in routine diagnostics [24], and it is not an exact test to determine phenotype. Among the 23 *mec*A-positive isolates from Argentina, two were oxacillin susceptible. In a *S. pseudintermedius* study carried out in Korea by Kang et al. [25], they found two isolates harboring the *mec*A gene but with susceptibility to methicillin and therefore classified as pre-MRSP. Pre-MRSP strains contain a functionally intact *mec*R1-*mec*I regulatory region. As a result, *mec*A expression might be strongly repressed. Under antibiotic pressure, such pre-MRSPs tend to constitutively produce penicillin-binding protein (PBP2a) and acquire increased resistance to methicillin. Both isolates from Argentina harbored the *mec*R1 (signal traducer) and *mec*I (methicillin-resistant repressor) regulators. In a study carried out in the U.S., Bruce et al. [26] also found discrepancies between the in vitro phenotypic testing for methicillin resistance and the in silico detection of the *mec*A gene [26].

Using an automated antimicrobial susceptibility test system, we could obtain high sensitivity and few discrepancies when comparing phenotype to genotype, resulting in an overall correlation of 96.3% in resistance phenotypes-genotypes. Recently, Tyson et al. [27] stated that whole-genome sequencing (WGS) could be a future efficient tool for antimicrobial resistance identification in *S. pseudintermedius*. The most common antimicrobial resistance genes found for the MRSP phenotypes are the *bla*Z, *erm*B, *aac*(6)-Ie-*aph*(2′)-Ia, *aph*(3′)-IIIa, *ant*(6′)-Ia, *tet*M, and *dfr*G genes [20,27,28].

In the present study, all isolates resistant to trimethoprim/sulfamethoxazole carried the *dfr*G gene. For gentamicin, six isolates from Argentina harbored the *aac*(6′)-*aph*(2″)-Ia gene that encodes a bi-functional enzyme, but only three of these isolates showed the resistant phenotype to gentamicin. The *aac*(6′)-*aph*(2″)-Ia gene was reported as the most frequent aminoglycoside modifying enzyme in *Staphylococcus* spp. [29]. Among the U.S. isolates, 11 carried the genes but only 5 were phenotypically resistant. Tyson et al. [27] also found discrepancies with gentamicin, and they observed that only 60% of gentamicin-resistant isolates carried the gene, whereas 17.4% of gentamicin-susceptible isolates were positive for this gene. In with a study by Perreten et al. [28], isolates containing the *aac*(6′)–*aph*(2″)-Ia gene displayed MICs of either 4 or 8 mg/L for gentamicin, for which they were considered borderline susceptible or intermediate, and they concluded the presence/absence of the gene is not a determining factor, but rather it is the expression level and/or copy number of the gene that may influence the phenotype.

Resistance to tetracycline in *S. pseudintermedius* is due mainly to the *tet*M gene, followed by *tet*K [30]. For the isolates from Argentina, all of those that were tetracycline resistant (*n* = 5) harbored the *tet*M gene, whereas two isolates from the U.S. carried both the *tet*M and *tet*K genes. These two genes encode different resistance mechanisms for tetracyclines: *tet*M codes for ribosome protective proteins, whereas *tet*K codes for efflux pumps [31]. Tetracycline resistance attributed to both genes (*tet*M or *tet*K) was reported previously by Smith et al. [20] in isolates from the U.S. Additionally, the presence of *tet*K was previously associated with the SCC*mec* Vc element [32] and the SCC*mec* of these two isolates were classified as Vc in our study as well.

In this study, the lincosamide nucleotidyltransferase gene *lnu*A that confers resistance only to lincosamides, but not to macrolides [16], was only detected in one U.S. isolate but none from Argentina.

Some studies have shown that isolates containing the genes *aph*(3’)-IIIa, *ant*(6)-Ia and *sat*4, also harbor the *erm*B gene [16,28]. In our study, 91.7% of the isolates from the U.S. that were positive to *aph*(3’)-IIIa:*ant*(6)-Ia:*sat*4 also carried the *erm*B, but in the isolates from Argentina only 68.2% of *aph*(3’)-IIIa:*ant*(6)-Ia:*sat*4-positive isolates also carried the *erm*B gene.

DNA topoisomerase IV is the main target of fluoroquinolones in *S. aureus,* and fluoroquinolone resistance results from mutations in the quinolone resistance-determining region of *gyr*A and *grl*A [33]. We detected isolates resistant to fluoroquinolones if they harbored two point mutations: *grl*A(S80I)-*gyr*A(S84L) or *grl*A(D84N)-*gyr*A(S84L). Only one isolate from the U.S. harbored the latter. Mutations in the *grl*A gene without additional *gyr*A mutations have been described for fluoroquinolone-resistant *S. aureus* strains and Descloux et al. [9] showed that *S. pseudintermedius* strains harboring at least one of these mutations presented decreased susceptibility to enrofloxacin, and higher MICs were observed when two additional mutations were present in *gyr*A. In this study, five *S. pseudintermedius* isolates with intermediate susceptibility to enrofloxacin showed a single mutation in *gyl*A(S80I). Among isolates from the U.S., a single mutation in *gyr*A was detected in four, two of which showed susceptibility and two showed intermediate susceptibility to enrofloxacin.

Mutations in the *rpo*B gene that involve a single amino acid change cause resistance to rifampin in MRSP [34]. Here, only two Argentine isolates were rifampin-resistant and harbored the mutation *rpo*B (H481N). This mutation was reported to be the most frequent in rifampin-resistant *S. aureus* isolates [35], but different amino acid positions have been observed for *S. pseudintermedius* [27,34].

Resistance against chloramphenicol was detected in four isolates from the U.S. and nine from Argentina. For *Staphylococcus* spp., the gene commonly associated with resistance to this antimicrobial is the *cat*A gene, a plasmid-borne chloramphenicol transferase [36]. In our study, most of the isolates that showed resistance to chloramphenicol were positive for *cat*A, but in three isolates from Argentina no genes were detected.

Using the SCC*mec*Finder Tool, SCC*mec* type prediction was successful for all the MRSP U.S. isolates, but only for 52.2% of the MRSP isolates from Argentina. As this tool is designed for *S. aureus*, and not all SCC*mec* elements described in *S. pseudintermedius* are included in the database, final SCC*mec* classification was obtained by aligning our isolates to known novel SCC*mec* references available in GenBank; however, more studies will be necessary to identify the SCC*mec* types not determined here, as the matches found were not 100% exact. In this regard, ensuring the quality of assemblies is essential for gene annotation/prediction because if assemblies are too fragmented, the predictions may be unreliable and alignments become cumbersome.

In this study, the three isolates predicted as SCC*mec* III(3A) by SCC*mec*Finder were mapped against the reference strains harboring the SCC*mec* II-III and SCC*mec* III, resulting in the highest homology to the *S. pseudintermedius* KM1381 reference strain [9], which harbors SCC*mec* II-III. This SCC*mec* is a hybrid of SCC*mec* II from *S. epidermidis* and SCC*mec* III from *S. aureus*, but with the cadmium operon absent. Inconsistencies in the nomenclature of SCC*mec* elements for *S. pseudintermedius* make it difficult to adequately compare studies, as this SCC*mec* element has also been referred to as “SCC*mec* type II-III” (hybrid), “ST71-SCC*mec* III”, or simply “SCC*mec* III”, which can lead to confusion between species. Recently, Bruce et al. [26] highlighted this need for a more standardized SCC*mec* nomenclature that captures more *Staphylococcus* species to more precisely assign SCC*mec* types and variants. Epidemic clones of sequence type 71-carrying SCC*mec* III were found for the first time in Argentina [37].

We also aligned/mapped the SCC*mec* V elements predicted by SCC*mec*Finder to the reference strains available. The mapping analysis revealed the presence of the *mec*R1 gene in five of the six U.S. isolates, and in only two out of 10 isolates from Argentina. Some of them had the *mec*R1 truncated, resulting in variable fragment sizes of this regulator. Similar findings were reported previously by Worthing et al. [12] in 10 *S. pseudintermedius* isolates, which they classified as SCC*mec* V_T_. Black et al. [38] first reported this truncated version in 37 *S. pseudintermedius* isolates that displayed homology to a *S. aureus* SCC*mec* V_T_ except for a deleted fragment of a gene.

In our study, the SCC*mec* type V isolates with one *ccr*C1 complex aligned with the *S. aureus* SCC*mec* Va (5C2) reference (AB121219), but they did not show high homology, which could suggest that these may be variants of SCC*mec* V (5C2). In a previous study carried out in Argentina with ten MRSP isolates [39], all isolates were identified as variants of SCC*mec* type V (5C2&5). Here, SCC*mec*Finder classified seven of the SCC*mec* cassettes from the Argentine isolates as SCC*mec* type V (5C2&5); these SCC*mec* V types with two *ccr*C1 were homologous to *S. pseudintermedius* 06-3228 (FJ544922.1) and *S. pseudintermedius* 23929 (ERR175868), which were classified as SCC*mec* V_T_ due to the truncated *mec*R1 gene, and showed similarity with the *S. aureus* SCC*mec* V_T_ [40], which was reclassified as VII [41], recently renamed as SCC*mec* Vb by Uehara [11].

Multi-locus sequence typing has identified several dominant MRSP clonal complexes (CC) around the world, including CC71 in Europe, CC68 in the United States, and CC45/CC112 in Asia [19]. CCs are groups of sequence types that share at least six identical alleles of a total of seven corresponding to the *S. pseudintermedius* MLST scheme, with strains then diverging from the predicted clonal ancestor [42]. In our analysis, no STs were a single-locus variant from CC45 (ST45) and CC71 (ST71). Rather, sequence types ST1412 and ST313 presented a double-locus variant with respect to ST45, and ST2250 presented a double-locus variant with respect to ST71. Phylogenetically, MSSP-ST2250 did not show a close phylogenetic relationship to other ST71 isolates. According to Pires dos Santos et al. [43], CC71 has only been detected in methicillin-resistant strains. Studies in Europe have reported CC71 as the current predominant clone in MRSP [19,44], but a recent study actually pointed to a decrease in the detection of MRSP belonging to ST71 in Europe [30]. In South America, ST71 was identified in Brazil in 2013 [45], and more recently, CC71 was found widely distributed throughout the country [46]. Furthermore, in a study that compiled information from 24 countries, MRSP CC71 showed resistance to trimethoprim/sulfamethoxazole in 70% of the isolates, resistance to tetracycline in 50% and resistance to chloramphenicol in 40% [43]. In the present study, antimicrobial resistance profiles of CC71 among Argentine and U.S. strains differed; Argentine strains were resistant to chloramphenicol and rifampin, whereas the U.S. strain was susceptible to both antimicrobials. Low incidence of resistance to chloramphenicol and tetracycline was reported previously for CC71 in the U.S. [47]. Resistance to tetracyclines varies in the global population of CC71 strains, with an average of 50% of strains resistant to this drug [43]. Here, the three isolates that belonged to ST71 did not show resistance to tetracycline and did not harbor any resistance genes associated to this antimicrobial.

Furthermore, associations have been made regarding sequence type and SCC*mec*. In this study, SCC*mec* type III was identified in all ST71 and in one ST440, in concordance with other reports that showed SCC*mec* III to be mainly associated with MRSP isolates belonging to ST71 [17,26,28].

Regarding the other dominant clone, ST45 among the MRSP isolates from Argentina presented the most extensive resistance profile, showing resistance to 18 of the 23 antimicrobials tested, but susceptible to trimethoprim/sulfamethoxazole, in contrast to the global population of strains belonging to this sequence type [43]. Additionally, while SCC*mec*Finder was not successful at typing the SCC*mec* cassettes of ST45 isolates, manual alignment/mapping could find homology to ΨSCC*mec*_57395_. Even though this association was found in here and in previous studies [12,48], it is not always the case [49,50].

With respect to SCC*mec* V, it was detected in different STs. Among the seven ST339 identified in Argentina, only two were classified as SCC*mec* type V(5C2) and the rest were non-typeable, which is not uncommon [49]. Regardless, this ST has been reported previously in MRSP from canine pyoderma in Argentina [39] and the Netherlands [51].

The only isolate with a SCC*mec* VII variant in this study was associated with a new ST first described here (ST2263).

Although the sample size for this study was not large (*n* = 29), we could observe that some MRSP shared the same ST and some belonged to dominant clones (CC71 and CC45), in concordance with the global trend, for which MRSP tend to be associated with a limited number of clones and MSSP tend to be more genetically diverse [43]. This is the first study to extensively analyze the population structure of *S. pseudintermedius* in the Buenos Aires Metropolitan Area in Argentina.

Argentine isolates showed the highest antimicrobial resistance for all the antimicrobials tested, with the exception of tetracycline, which was higher among U.S. isolates. This could be that the consequence of the choice of treatment for pyodermas in Argentina, which can be topical and/or systemic, where antimicrobials from the group of cephalosporins, lincosamides, and fluoroquinolones are the most commonly used. From the population structure analysis, we found that the U.S. isolates showed genetic diversity with the absence of a prevalent clone, whereas among the MRSP isolates from Argentina, ST339 was the most prevalent clone, previously reported in only one isolate from Argentina. In addition, we detected two ST71 and one ST45 among Argentine isolates, which belong to the dominant MRSP clonal complexes that have spread in Europe and Asia, respectively, and more recently have attained global distribution. A more extensive sampling of *S. pseudintermedius* in Argentina, in terms of geographical origin, will be necessary in order to determine the population of this organism in other parts of the country. The continued identification and characterization of *S. pseudintermedius* is important to understand the epidemiology and clonal relationships of MRSP strains, which are of great importance in both veterinary and human medicine.

## 4. Materials and Methods

### 4.1. Sample Collection

*S. pseudintermedius* isolates from dogs with pyoderma were collected in 2016 from the Buenos Aires Metropolitan Area (Ciudad Autónoma de Buenos Aires, Gran Buenos Aires and La Plata, Argentina). Thirty isolates were selected at random from this strain collection, housed in the Laboratory of Bacteriology and Antimicrobials (Department of Microbiology, Faculty of Veterinary Sciences, Universidad Nacional de La Plata [UNLP]) and subjected to biochemical testing for initial identification as *S. pseudintermedius*. A paired cohort of *S. pseudintermedius* isolates recovered from dogs with pyoderma collected in 2019–2021 throughout the U.S. were selected at random from the National Animal Health Laboratory Network (NAHLN) Antimicrobial Resistance Pilot Project strain collection, housed at the NVSL in Ames, IA, USA.

Confirmatory identification of all isolates was performed through matrix-assisted laser desorption/ionization and qualitative time of flight (MALDI-qTOF) and sequence-based alignment using Kraken’s default parameters [52] at 90% reference genome coverage or greater against a customized database consisting of RefSeq complete genomes database release ver. 209 [53], UniVec-core, and host genomes commonly encountered by the NVSL. One isolate was identified as *S. schleiferi* and was excluded, alongside its matched pair, from this study for a total of *n =* 29 for each sample population.

### 4.2. Antimicrobial Susceptibility Testing, Heatmaps, and Correlations

To determine phenotypic resistance profiles, antimicrobial susceptibility testing (AST) was performed on all MRSP clones using the Sensititre™ COMPGP1F plate (Thermo Fisher Scientific, Cleveland, OH, USA) followed by CLSI interpretations as outlined in the *Vet01S* [21]. For some antimicrobial agents, clinical breakpoints were unavailable in the *Vet01S* for dogs; in these cases, human breakpoint values were used as the cutoff for resistance. Verification for the presence of the *mec*A gene was performed using WGS (described below). In cases of discordance, in vitro AST were repeated.

Dendrograms were generated in R (http://www.R-project.org/, accessed on 13 June 2022) using hierarchical clustering in the base *stats* package on Euclidean distance matrices representing ordinal AMR interpretations for each antimicrobial class and isolate (where 0 = susceptible, 1 = intermediate, 2 = resistant). Heatmaps were generated in R with *ggplot2* [54] with the same ordinal AMR interpretation data for each antimicrobial class and isolate.

For phenotype-to-genotype correlations, minimum inhibitory concentration (MIC) values below the CLSI-provided resistance breakpoint for each antimicrobial drug were considered susceptible for genotype–phenotype correlations, including MIC values that would otherwise be considered intermediate interpretations. Genotype and phenotype results were then correlated such that isolates with phenotypic resistance to an antimicrobial identified by AST also had an associated resistance gene or mutation identified by WGS, or isolates with phenotypic susceptibility to an antimicrobial also lacked any associated resistance gene or mutation.

### 4.3. Whole-Genome Sequencing

DNA was extracted and used to prepare indexed genomic libraries using the Nextera XT^®^ DNA Library Preparation Kit (Illumina Corp., San Diego, CA, USA). Multiplexed libraries were sequenced on an Illumina MiSeq^®^ platform using 2 × 250 paired end read chemistry. All raw sequence data in *.fastq.gz format were deposited in the National Center for Biotechnology Information (NCBI) under BioProjects PRJNA848756 (Argentina) and PRJNA510385 (United States).

### 4.4. Population Structure Analysis

Multi-locus sequence typing (MLST) was determined using ABRicate based on the *S. pseudintermedius* MLST database, and new alleles and STs were submitted to PubMLST (http://pubmlst.org/spseudintermedius, accessed on 23 March 2022) for curation and ST number designation by Vincent Perreten (vincent.perreten@vetsuisse.unibe.ch). Briefly, all STs were grouped using the BURST tool on the PubMLST website [55], assigned to clonal complexes (CCs) using goeBURST [56], and illustrated with Phyloviz v 2.0 (http://phyloviz.net/, accessed on 12 May 2022, [57]). Clonal complexes were defined as any group of strains sharing 5+ identical profile alleles, using the double-loci variants (DVL) parameter.

Phylogenetic reconstruction was performed using the concatenated single-nucleotide polymorphism (SNP) sequences obtained from mapping the isolates’ whole-genome sequences against the reference genome *Staphylococcus pseudintermedius* HKU10-03 (GenBank accession number NC014925) using the vSNP pipeline (https://github.com/USDA-VS/vSNP, accessed on 29 November 2021). This pipeline uses short read alignment to the reference using BWA-MEM [58] followed by SNP calling with FreeBayes [59]. Alignment of calls across the SNP positions were then used to build a phylogenetic tree using RAxML [60] under a maximum-likelihood algorithm and general time reversible (GTR-CATI) model. Tree editing, annotation and visualization was performed with iTOL (https://itol.embl.de/, accessed on 12 May 2022).

### 4.5. Antimicrobial Genomic Analysis and SCCmec Typing

To determine the presence of AMR genes and fluoroquinolone point mutations, all sequences were de novo assembled with Spades ver. 3.14.0 [61], and resultant scaffolds were analyzed against AMRFinder Plus [62] using the NCBI database and ABRicate (https://github.com/tseemann/abricate, accessed on 5 May 2022) using the ResFinder database applying an identity threshold of >80% and a minimum coverage of ≥60% of the target gene. For rifampicin resistance, the *rpo*B gene sequence was extracted from each isolate by mapping the reference sequence (GenBank accession number CP002478.1:2305191-2308745) to each of the genomes and a subsequent alignment to identify the mutation site using Geneious Prime v11.0.9+11 (Biomatters Ltd., Auckland, NZ).

SCC*mec* types were determined using SCC*mec*Finder version 1.2 (https://cge.food.dtu.dk/services/SCCmecFinder-1.2/, accessed on 21 April 2022), which uses a database that compiles all the reference SCC*mec* types (I through XII) and subtypes (IV through V) listed by the IWG-SCC. In cases where SCC*mec* types III and V were identified, we performed additional manual alignment/mapping of the isolates to the available reference sequences for these SCC*mec* types, including sequences from *S. pseudintermedius* (AB03671.1, AM904732.1 for SCC*mec* type III and II-III, respectively, and FJ544922.1, ERR175868, AB512767.1, AB505629.1, AB478780.1, AB462393.1, AB121219.1 for SCC*mec* type V), using Geneious Prime v11.0.9+11 (Biomatters Ltd). In cases where SCC*mec* type IV was identified, we performed additional manual alignment/mapping of all the SCC*mec* type IV subtypes described by Uehara [11] (AB063172.1, AB063173.1, AB096217.1, AB097677.1, DQ106887.1, HE681097.1, AB425823.1, AB425824.1, GU122149.1, AB633329.1, AB872254.1, KX385846.1). In cases where SCC*mec* type VII was identified, we performed a comparative analysis using the *S. aureus* reference for SCC*mec* VII (AB373032.1) against SCC*mec* NA45 (CP016072.1), a novel SCC*mec* VII variant in *S. pseudintermedius* with an inverted *mec* gene [12]. The best match was determined by using the highest homology (ID%) and coverage (%).

## Figures and Tables

**Figure 1 ijms-24-11361-f001:**
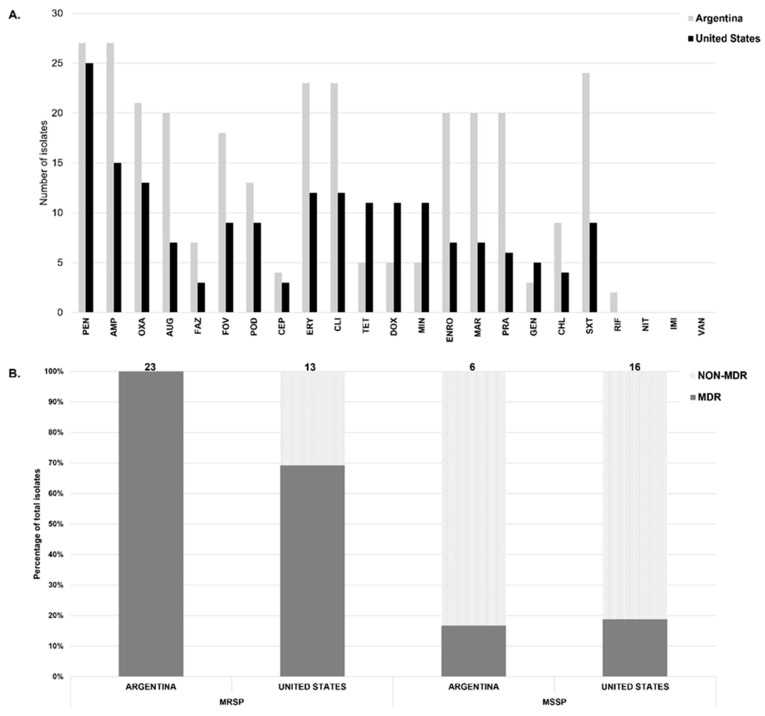
(**A**) Counts of *S. pseudintermedius* isolates resistant to each antimicrobial in dogs with pyoderma from Argentina (grey) and the United States (black). Antibiotics: PEN (penicillin), AMP (ampicillin), OXA (oxacillin), AUG (amoxicillin/clavulanate), FAZ (cefazolin), FOV (cefovecin), POD (cefpodoxime), CEP (cephalothin), ERY (erythromycin), CLI (clindamycin), TET (tetracycline), DOX (doxycycline), MIN (minocycline), ENRO (enrofloxacin), MAR (marbofloxacin), PRA (pradofloxacin), GEN (gentamicin), CHL (chloramphenicol), SXT (trimethoprim/sulphamethoxazole), RIF (rifampin), NIT (nitrofurantoin), IMI (imipenem), and VAN (vancomycin). (**B**) Relative proportions of methicillin-resistant S. pseudintermedius (MRSP) and methicillin susceptible (MSSP) in each country, shown here as either multi-drug resistant (MDR, grey) or non-MDR (light grey).

**Figure 2 ijms-24-11361-f002:**
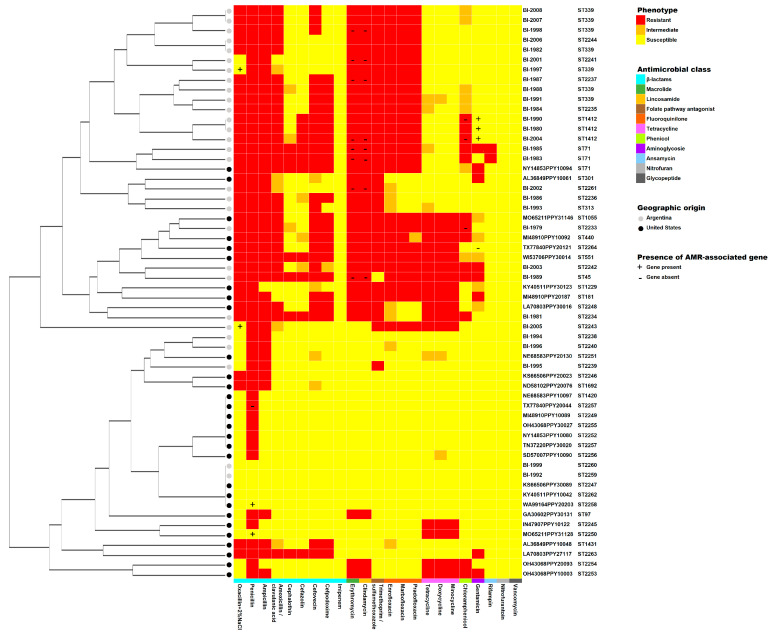
**Heatmap and dendrogram of the antimicrobial resistance profiles of *S. pseudintermedius* isolates from dogs with pyoderma from Argentina and the United States**. The dendrogram was generated based on similarity of profiles using a hierarchical clustering approach on ordinal resistance interpretation data (where 0 = susceptible, 1 = intermediate, and 2 = resistant). Phenotypes were classified as resistant (red), intermediate (orange) and susceptible (yellow) based on Sensititre™ results; the antimicrobials tested are indicated on the x-axis and are grouped by antimicrobial class according to the legend. Geographic origin (Argentina–grey circle; United States–black circle), sequence type (ST), and AMR-associated gene presence/absence (+/−) are indicated.

**Figure 3 ijms-24-11361-f003:**
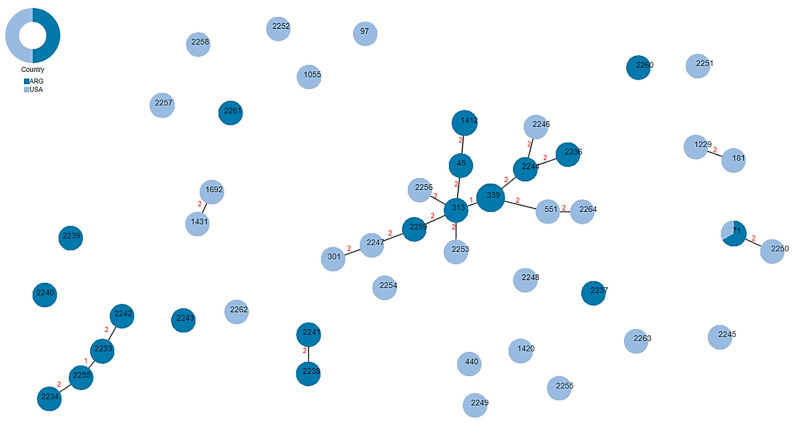
**Clonal relationships of *S. pseudintermedius* isolates from Argentina (dark blue) and the United States (light blue) based on MLST profiles**. Minimum spanning tree (MST) was constructed with the *goe*BURST algorithm in PHYLOViZ (http://phyloviz.net/, accessed on 12 May 2022) with a tree cut-off value = 3 (allelic profiles with five or more allele matches). STs are represented by each node and the size of the node represents number of isolates identified with that ST. Links represent the number of locus variant differences in ST between nodes (i.e., 1 = one locus variant, 2 = locus variants) and singletons are represented by nodes with no connections.

**Figure 4 ijms-24-11361-f004:**
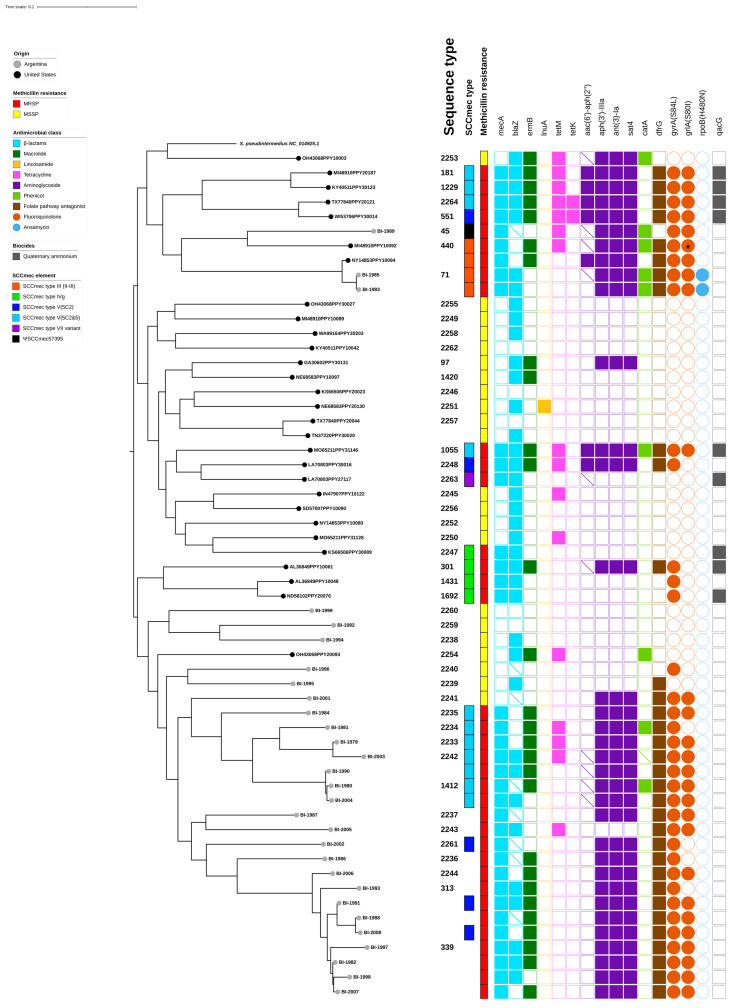
**Single-nucleotide polymorphism (SNP) phylogeny-based comparison of antimicrobial resistance (AMR) and biocide-associated genes observed in *S. pseudintermedius* isolates from canine pyoderma cases in Argentina (grey tree tips) and the United States (black tree tips**). Genotyping characteristics are shown next to the midpoint-rooted SNP phylogeny tree, showing data for each isolate in a heat map. Heatmap colors and shapes correspond to SCC*mec* type, methicillin resistance (based on *mec*A detection), and antimicrobial classes associated with each AMR and biocide-associated gene as indicated in the legend. Partial gene detections (query gene had <60% of the length of the target reference gene) are indicated with a diagonal marking. The asterisk (*) represents a variant mutation in *grl*A that corresponds to a change from Aspartate to Asparagine (D84N). Tree editing, annotation and visualization was performed with iTOL.

**Figure 5 ijms-24-11361-f005:**
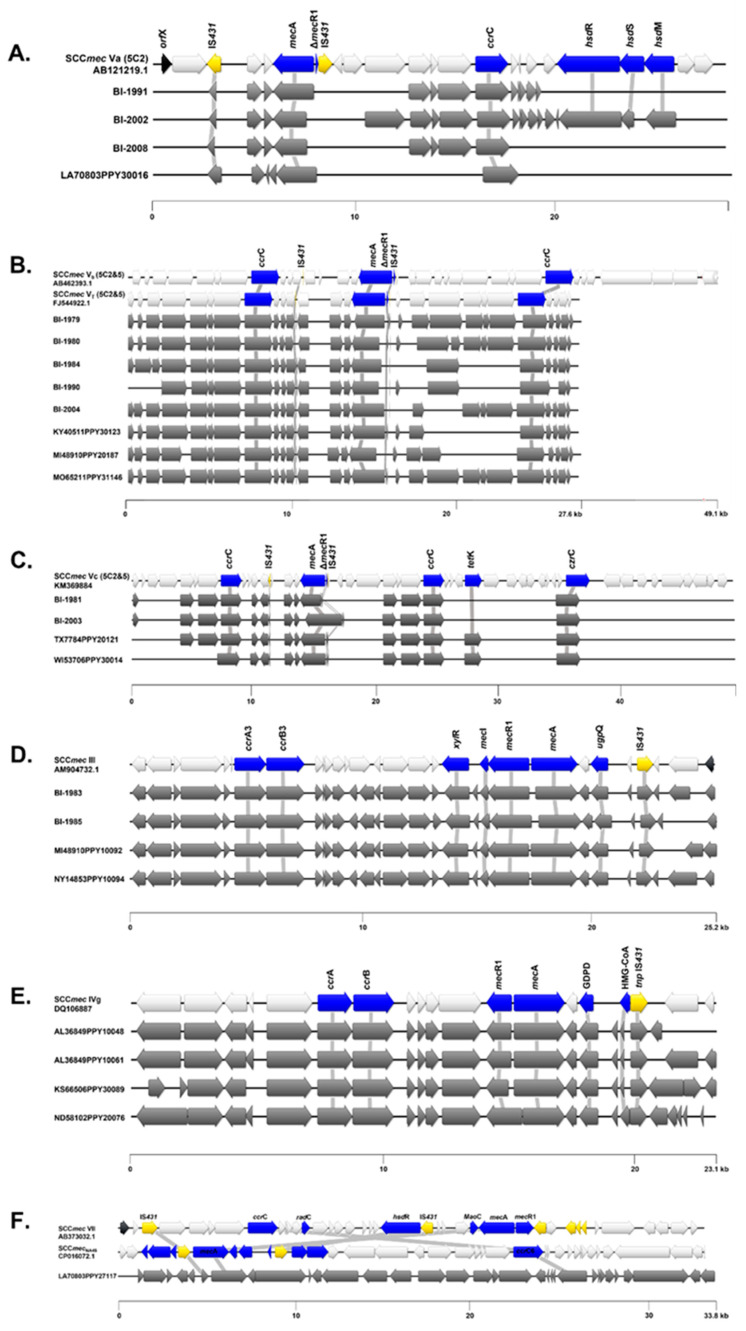
**Comparison of the predicted SCC*mec* cassettes against their corresponding references.** (**A**) SCC*mec* type Va (5C2); (**B**) SCC*mec* type Vb (5C2&5); (**C**) SCC*mec* type Vc (5C2&5); (**D**) SCC*mec* type III (II–III); (**E**) SCC*mec* type IVg; (**F**) SCC*mec* type VII and SCC*mec* VII variant (SCC*mec*_NA45_). In panel F, the U.S. isolate shown here shared the greatest homology to SCC*mec_NA45_*, but presented *ccr*C6 in a juxtaposed position, for which it may represent a novel variant of this SCC*mec* type. Only SCC*mec* components (*mec*A, *ccr*, and complementary genes; blue) and insertion sequences/mobile elements (yellow) are highlighted. Elements annotated as hypothetical proteins are white. Grey lines indicate correspondence of SCC*mec* components between the isolate(s) in question and the respective references.

**Table 1 ijms-24-11361-t001:** Sequence types (ST) identified by multi-locus sequence typing (MLST) for the *S. pseudintermedius* isolates, which have been previously described.

ST	Argentina	United States
45	*n* = 1	
71	*n* = 2	*n* = 1
97		*n* = 1
181		*n* = 1
301		*n* = 1
313	*n* = 1	
339	*n* = 1	
440		*n* = 1
551		*n* = 1
1055		*n* = 1
1229		*n* = 1
1412	*n* = 3	
1420		*n* = 1
1431		*n* = 1
1692		*n* = 1

**Table 2 ijms-24-11361-t002:** Genotype–phenotype correlations among *n* = 58 *S. pseudintermedius* isolates for sixteen antimicrobials.

Antimicrobials	Phenotype: S		Phenotype: R		
	Genotype: R	Genotype: S	Genotype: R	Genotype: S	Correlation
Oxacillin	2	22	34	0	96.5%
Penicillin	2	4	51	1	94.8%
Erythromycin	0	23	27	8	86.2%
Clindamycin	0	23	27	8	86.2%
Trimethoprim-Sulfamethoxazole	0	25	33	0	100%
Enrofloxacin	0	31	27	0	100%
Marbofloxacin	0	31	27	0	100%
Pradofloxacin	1	30	27	0	98.3%
Tetracycline	0	42	16	0	100%
Doxycycline	0	42	16	0	100%
Minocycline	0	42	16	0	100%
Chloramphenicol	0	45	10	3	94.8%
Gentamycin	9	41	8	0	84.5%
Rifampin	0	56	2	0	100%
Nitrofurantoin	0	58	0	0	100%
Vancomycin	0	58	0	0	100%

S = susceptible, R = resistant. Analysis of correlation was performed using the calculation: isolates with concordant phenotypes (S:S or R:R)/all isolates tested against an antimicrobial × 100.

**Table 3 ijms-24-11361-t003:** **SCC*mec* cassette characterization of *S. pseudintermedius* isolates from Argentina and the United States.** ^1^ Detected with ABRicate (ResFinder database); ^2^ SCC*mec* types were confirmed through manual inspection and mapping of the SCC*mec* reference sequences to the genomes; n/d = not detected; R = resistant; S = susceptible; (-) = SCC*mec* or MLST match not found.

Isolate ID	Source	Oxacillin MIC (µg/mL)	Interpretation	SCC*mec*Finder					Best SCC*mec* Match ^2^	MLST
				*mec* Class	*mec* Complex Genes	*ccr* Class	*ccr* Complex Genes	Predicted SCC*mec*		
BI-1979	Argentina	2	R	C2	*mec*A	5&5	*ccr*C1-allele-8, *ccr*C1-allele-3	SCC*mec* type V(5C2&5)	SCC*mec* type Vb (5C2&5)	2233
BI-1980	Argentina	>2	R	C2	*mec*A	5&5	*ccr*C1-allele-2, *ccr*C1-allele-8	SCC*mec* type V(5C2&5)	SCC*mec* type Vb (5C2&5)	1412
BI-1981	Argentina	>2	R	C2	*mec*A	5&5	ccrC1-allele-8, ccrC1-allele-3	SCC*mec* type V(5C2&5)	SCC*mec* type Vc (5C2&5)	2234
BI-1982	Argentina	2	R	n/d	*mec*A, *mec*I, Δ*mec*R1	5	*ccr*C1-allele-8	-	-	-
BI-1983	Argentina	>2	R	A	*mec*A, *mec*I, *mec*R1	3	*ccr*A3, *ccr*B3	SCC*mec* type III(3A)	SCC*mec* type III	71
BI-1984	Argentina	>2	R	C2	*mec*A	5&5	*ccr*C1-allele-8, *ccr*C1-allele-1	SCC*mec* type V(5C2&5)	SCC*mec* type V (5C2&5)	2235
BI-1985	Argentina	>2	R	A	*mec*A, *mec*I, Δ*mec*R1	3	*ccr*A3, *ccr*B3	SCC*mec* type III(3A)	SCC*mec* type III	71
BI-1986	Argentina	>2	R	n/d	*mec*A, *mec*I, Δ*mec*R1	5	*ccr*C1-allele-8	-	-	-
BI-1987	Argentina	>2	R	A	*mec*A, *mec*I, *mec*R1	5	*ccr*C1-allele-8	-	-	-
BI-1988	Argentina	>2	R	n/d	*mec*A, *mec*I, Δ*mec*R1	5	*ccr*C1-allele-8	-	-	-
BI-1989	Argentina	>2	R	n/d	*mec*A	n/d		-	ψSCC*mec*_57395_	45
BI-1990	Argentina	>2	R	C2	*mec*A	5&5	*ccr*C1-allele-8, *ccr*C1-alelle-3	SCC*mec* type V(5C2&5)	SCC*mec* type Vb (5C2&5)	1412
BI-1991	Argentina	>2	R	C2	*mec*A	5	*ccr*C1-allele-8	SCC*mec* type V(5C2)	SCC*mec* type Va (5C2)	339
BI-1993	Argentina	1	R	n/d	*mec*A, Δ*mec*R1	5	*ccr*C1-allele-8	-	-	-
BI-1997	Argentina	≤0.25	S	n/d	*mec*A, *mec*I, Δ*mec*R1	5	*ccr*C1-allele-8	-	-	-
BI-1998	Argentina	2	R	n/d	*mec*A, *mec*I, Δ*mec*R1	5	*ccr*C1-allele-8	-	-	-
BI-2002	Argentina	0.5	R	C2	*mec*A	5	*ccr*C1-allele-8	SCC*mec* type V(5C2)	SCC*mec* type Va (5C2)	2261
BI-2003	Argentina	2	R	n/d	*mec*A ^1^	5&5	*ccr*C1-allele-3	SCC*mec* type V(5C2&5)	SCC*mec* type Vc (5C2&5)	2242
BI-2004	Argentina	>2	R	C2	*mec*A	5&5	*ccr*C1-allele-2, *ccr*C1-allele-8	SCC*mec* type V(5C2&5)	SCC*mec* type Vb (5C2&5)	-
BI-2005	Argentina	≤0.25	S	n/d	*mec*A, *mec*I, Δ*mec*R1	5	*ccr*C1-allele-8	-	-	-
BI-2006	Argentina	2	R	n/d	*mec*A, Δ*mec*R1	5	*ccr*C1-allele-8	-	-	-
BI-2007	Argentina	1	R	n/d	*mec*A	n/d		-	-	-
BI-2008	Argentina	>2	R	C2	*mec*A	5	*ccr*C1-allele-8	SCC*mec* type V(5C2)	SCC*mec* type Va (5C2)	339
AL36849PPY10048	United States	1	R	B	*mec*A, IS*1272*, Δ*mec*R1	2	*ccr*A2, *ccr*B2	SCC*mec* type IVg(2B)	SCC*mec* type IVg	1431
AL36849PPY10061	United States	0.5	R	B	*mec*A, IS*1272*, Δ*mec*R1	2	*ccr*A2, *ccr*B2	SCC*mec* type IVg(2B)	SCC*mec* type IVg	301
KS66506PPY30089	United States	0.5	R	n/d	*mec*A, IS*1272*, Δ*mec*R1	2	*ccr*A2, *ccr*B2	SCC*mec* type IV(2B)	SCC*mec* type IV	2247
KY40511PPY30123	United States	0.5	R	C2	*mec*A	5	*ccr*C1-allele-2, *ccr*C1-allele-8	SCC*mec* type V(5C2&5)	SCC*mec* type Vb (5C2&5)	1229
LA70803PPY27117	United States	>2	R	C1	*mec*A	5	*ccr*C1-allele-6	SCC*mec* type VII(5C1)	SCC*mec* type VII variant	2263
LA70803PPY30016	United States	2	R	C2	*mec*A	5	*ccr*C1-allele-6	SCC*mec* type V(5C2)	SCC*mec* type Va (5C2)	2248
MI48910PPY10092	United States	>2	R	n/d	*mec*A, *mec*I, *mec*R1	3	*ccr*A3, *ccr*B3	SCC*mec* type III(3A)	SCC*mec* type III	440
MI48910PPY20187	United States	1	R	C2	*mec*A	5	*ccr*C1, *ccr*C1-allele-3	SCC*mec* type V(5C2&5)	SCC*mec* type Vb (5C2&5)	181
MO65211PPY31146	United States	>2	R	C2	*mec*A	5	*ccr*C1-allele-2, *ccr*C1-allele-8	SCC*mec* type V(5C2&5)	SCC*mec* type Vb (5C2&5)	1050
ND58102PPY20076	United States	0.5	R	n/d	*mec*A ^1^, Δ*mec*R1, IS*1272*	2	*ccr*A2, *ccr*B2	SCC*mec* type IVg(2B)	SCC*mec* type IVg	1692
NY14853PPY10094	United States	>2	R	n/d	*mec*A, *mec*I, *mec*R1	3	*ccr*A3, *ccr*B3	SCC*mec* type III(3A)	SCC*mec* type III	71
TX77840PPY20121	United States	>2	R	C2	*mec*A	5&5	*ccr*C1-allele-2, *ccr*C1-alelle-8	SCC*mec* type V(5C2&5)	SCC*mec* type Vc (5C2&5)	2264
WI53706PPY30014	United States	>2	R	C2	*mec*A	5&5	*ccr*C1-allele-2, *ccr*C1-allele-8	SCC*mec* type V(5C2&5)	SCC*mec* type Vc (5C2&5)	551

## Data Availability

All raw sequence data in *.fastq.gz format were deposited in the National Center for Biotechnology Information (NCBI) under BioProjects PRJNA848756 (Argentina) and PRJNA510385 (United States).

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
