# Peer review of "Genomic Features of Antimicrobial Resistance in Staphylococcus pseudintermedius Isolated from Dogs with Pyoderma in Argentina and the United States: A Comparative Study"

_ijms, 2023, doi:10.3390/ijms241411361_

Round 1

Reviewer 1 Report

Please insert citations correctly in the text 

Lines 87-88 Please better explain the correlation between genotype, phenotype and TS

Lines  99-100 Please explain how the analysis of resistance genes was done; it is not clear in the text whether whole genome sequencing of the isolates was performed

Line 223 Please also specify in the text the software used to make the philogenetic trees

Author Response

Please insert citations correctly in the text 

References were corrected.

Lines 87-88 Please better explain the correlation between genotype, phenotype, and ST.

This phrase was edited.

Lines 99-100 Please explain how the analysis of resistance genes was done; it is not clear in the text whether whole genome sequencing of the isolates was performed.

It was added.

Line 223 Please also specify in the text the software used to make the phylogenetic trees.

It was added.

Reviewer 2 Report

The authors investigated the antimicrobial resistance features and the phylogeny of Staphylococcus pseudointermedius, particularly they aimed to comparatively analyze the population structure of canine pyoderma-associated S. pseudintermedius isolates in Argentina and in the United States.

The scope of this paper is in line with those of International Journal of Molecular Sciences and the analysis performed are scientifically sound. However, I would recommend that the authors carefully revise the points/suggestions below:

1.     Line 33: “it’s” should be “it is”.

2.     Line 142: MIC unit is missing.

3.     Line 446: “WGS” should be “Whole genome sequencing (WGS)”.

4.    Figure1: Only abbreviations of antibiotics were displayed, please put the full name of the antibiotics in the legend of the figure.

5. Discussion: I would recommend softening a bit the discussion and simplifying.

Author Response

Thanks for your suggestions.

  1. Line 33: “it’s” should be “it is”.

It was edited.

  1. Line 142: MIC unit is missing.

It was added.

  1. Line 446: “WGS” should be “Whole genome sequencing (WGS)”.

It was edited.

  1. Figure1: Only abbreviations of antibiotics were displayed, please put the full name of the antibiotics in the legend of the figure.

It was added

  1. Discussion: I would recommend softening a bit the discussion and simplifying.

It was shortened.